# Nuclear-driven production of renewable fuel additives from waste organics

Arran George Plant [1], Bor Kos[2], Anže Jazbec[2], Luka Snoj [2], Vesna Najdanovic-Visak [3✉] & Malcolm John Joyce [1✉]

Non-intermittent, low-carbon energy from nuclear or biofuels is integral to many strategies to achieve Carbon Budget Reduction targets. However, nuclear plants have high, upfront costs and biodiesel manufacture produces waste glycerol with few secondary uses. Combining these technologies, to precipitate valuable feedstocks from waste glycerol using ionizing radiation, could diversify nuclear energy use whilst valorizing biodiesel waste. Here, we demonstrate solketal (2,2-dimethyl-1,3-dioxolane-4-yl) and acetol (1-hydroxypropan-2-one) production is enhanced in selected aqueous glycerol-acetone mixtures with $\gamma$ radiation with yields of $1.5 \pm 0.2$ µmol $J^{-1}$ and $1.8 \pm 0.2$ µmol $J^{-1}$, respectively. This is consistent with the generation of either the stabilized, protonated glycerol cation ($CH_2OH\text{-}CHOH\text{-}CH_2OH_2^+$) from the direct action of glycerol, or the hydronium species, $H_3O^+$, via water radiolysis, and their role in the subsequent acid-catalyzed mechanisms for acetol and solketal production. Scaled to a hypothetically compatible range of nuclear facilities in Europe (i.e., contemporary Pressurised Water Reactor designs or spent nuclear fuel stores), we estimate annual solketal production at approximately $(1.0 \pm 0.1) \times 10^4$ t $year^{-1}$. Given a forecast increase of 5% to 20% v/v% in the renewable proportion of commercial petroleum blends by 2030, nuclear-driven, biomass-derived solketal could contribute towards net-zero emissions targets, combining low-carbon co-generation and co-production.

[1] Engineering Department, Lancaster University, Lancaster, UK. [2] Jožef Stefan Institute, Ljubljana, Slovenia. [3] Chemical Engineering and Applied Chemistry (CEAC), Energy & Bioproducts Research Institute (EBRI), Aston University, Birmingham, UK. ✉email: v.najdanovic@aston.ac.uk; m.joyce@lancaster.ac.uk

Nuclear power has the lowest carbon footprint[1] second only to wind and is not intermittent. However, the long periods separating upfront capital investments and generation revenues constitute a financial risk. According to the comprehensive techno-economic analysis published by Schmeda–Lopez[2], integration of a large nuclear power plant (NPP) and a chemical process using the reactor's γ radiation to facilitate the production of commodity chemicals such as propylene leads to significant economic benefits. This suggests an economically attractive route might exist to valorize waste biomass that avoids petrochemical production.

The effects of ionizing radiation on materials and media were researched extensively in the 20th century[3], realizing applications such as polymer synthesis and medical sterilization[4]. The efficiency of processes initiated by ionizing radiation in organic materials, i.e., radiolysis, is quantified by the radiation-chemical yield or G-value which is dependent on the target product, irradiation parameters and starting reagent, etc. For γ-ray radiolysis of organics, most radiation-chemical yields of stable products span[5] 0.1 to 1 µmol J$^{-1}$, with some exceptions for halogenated reagents. Relatively few radiolysis-based processes are used to synthesize chemicals at scale; an exception is the production of ethyl bromide with $^{60}$Co γ rays by the Dow Institute in 1963[6].

Of late, co-production with nuclear systems[2,7–10] has attracted attention, particularly for hydrogen production and water desalination. However, these options are comparable to electricity production in terms of profitability. An integrated chemical-nuclear process could be more economically favourable, and sympathetic with small modular reactor[11] and advanced (i.e., Generation IV) NPP designs[10] with relatively little additional capital cost[6], but few reports exist on the use of ionizing radiation to catalyze transformations in materials that produce valuable chemicals. Here, we propose the coupling of *nuclear* and *biorefinery* processes for the co-production of renewable fuel additives from waste glycerol.

Glycerol is produced as a by-product from biodiesel production but also has potential as a low-value source of valuable, renewable chemicals. Since the saturation of the glycerol market in 2006, due to the increase in biodiesel production[12,13], the price of glycerol whilst remaining relatively low has been rising steadily. Historically, glycerol has been unusable in high-value applications[14,15] with thousands of tonnes of crude glycerol being disposed of at negative prices in 2014[16]. As of 2017 in the EU, prices of crude and refined glycerol were at 200–300 € tonne$^{-1}$ and 500–700 € tonne$^{-1}$ (pre-pandemic)[17], respectively. With glycerol production expected to triple by 2030[18] and oversupply expected to continue, deriving useful feedstocks from waste glycerol is important if biodiesel production is to be sustainable.

Various catalysts have been used to convert glycerol to valuable chemicals[19,20], including acetol (a solvent and an intermediary used to produce polyols and acrolein, food flavouring and dyeing additive) and solketal (a fuel additive)[21,22]. However, catalytic conversion is often complicated by the deactivation of the catalyst, high temperature and pressure requirements, difficulty separating the catalyst from the product and long reaction times[23]. While chemoselective advances for acetol[24,25] and solketal[26] have been reported, radiation-initiated processing has not been explored for glycerol despite the potential to offer several advantages[27]: (i) catalytic deactivation or poisoning is not a concern; (ii) reactions can proceed at ambient temperatures and pressures; (iii) the availability of irradiating large reaction volumes due to the penetrating power of ionizing radiation; and (iv) the utilization of waste ionizing sources (spent nuclear fuel) would also result in negligible radiation-related processing costs.

Here, we present a radiolytic process that produces solketal, and which produces an enhanced yield of the previously known product, acetol, from neat and aqueous glycerol using γ and neutron radiations. Volumes and dose rates for hypothetical production environments (a Pressurized Water Reactor (PWR) and spent nuclear fuel pool) are simulated and combined with the G-value data to give the maximum annual production capacity of solketal and acetol for a hypothetical European nuclear-chemical production network.

## Results

**Irradiation of glycerol and its aqueous solutions**. Glycerol mixtures were irradiated by γ-ray only and mixed-field radiations (neutrons and γ rays) from a TRIGA Mark II reactor[28] producing various stable products (Supplementary Table 1). Acetol and solketal have been identified as major products. Figure 1a, b shows the corresponding radiation-chemical yields (G-values) from neat glycerol, as a function of absorbed dose for acetol and solketal, respectively, for each irradiation type. Figure 1c, d shows the G-values as a function of dose rate for 50 kGy mixed-field exposures for acetol and solketal, respectively. Additionally, Fig. 2a, b shows the radiation-chemical yields of acetol and solketal from binary aqueous glycerol and ternary glycerol, water and acetone mixtures for each radiation type, respectively. Furthermore, Fig. 2c, d shows the % molar yields from glycerol for acetol and solketal for a selection of irradiated glycerol mixtures compared against unirradiated control samples.

The data in Fig. 1a indicate a difference in yield for each radiation type, and contrasting dependencies of yield with absorbed dose. For acetol, γ-ray exposure results in a gradual decline in radiation-chemical yield with absorbed dose, whereas this increases with a dose for the mixed-field. Mixed-field yields are consistently less than γ-ray yields across the dose range, but the trend with dose implies a convergence with the γ-ray data. It is clear (Fig. 1b) that the γ-ray G-value response for solketal contrasts with that for acetol, with solketal yield increasing for both γ-ray and mixed-field exposures.

## Discussion

The influence of dose rate on yield for acetol is not conclusive from our data (Fig. 1c) but we note acetol yield is expected to decrease with dose rate based on prior art[29], as expanded upon below. The data in Fig. 1d suggest a similar decrease for solketal with increasing mixed-field dose rates from 500 to 8200 Gy min$^{-1}$. While the dose rates used in this research are much higher than those in the prior art (1.7–72 Gy min$^{-1}$), this research also demonstrates a decrease in yield for higher dose rates.

The data in Fig. 2 show irradiation of neat glycerol and binary (glycerol + water) produces a small yield of solketal. G-values are significantly higher when only γ radiation is used, as shown by a comparison between Fig. 2a, b. The addition of 69 mol% of water to glycerol provokes a significant increase in the yield of acetol by a factor of 3 compared to neat glycerol for both types of irradiation, from 0.6 ± 0.07 µmol J$^{-1}$ to a maximum of 1.8 ± 0.3 µmol J$^{-1}$. However, further dilution leads to a continuous decrease in acetol yield which compares well with data from previous reports[29] which quote a G-value of 0.22 µmol J$^{-1}$ to dilute aqueous concentrations of 0.93%mol of glycerol. Adding acetone to aqueous glycerol mixtures results in an increase in solketal yield by a factor of ~34 for γ-ray irradiations, from 0.045 ± 0.005 µmol J$^{-1}$ to a maximum of 1.53 ± 0.2 µmol J$^{-1}$. Whilst equivalent mixed-field irradiations produced only a relatively low increase in % molar yield of solketal, when compared with unirradiated controls, γ-irradiations resulted in a factor of 12 increase in yield.

Besides acetol and solketal, acetic acid was also detected in the ternary (glycerol + acetone + water) samples, consistent with acetone radiolysis[30] with maximum concentrations and radiation-

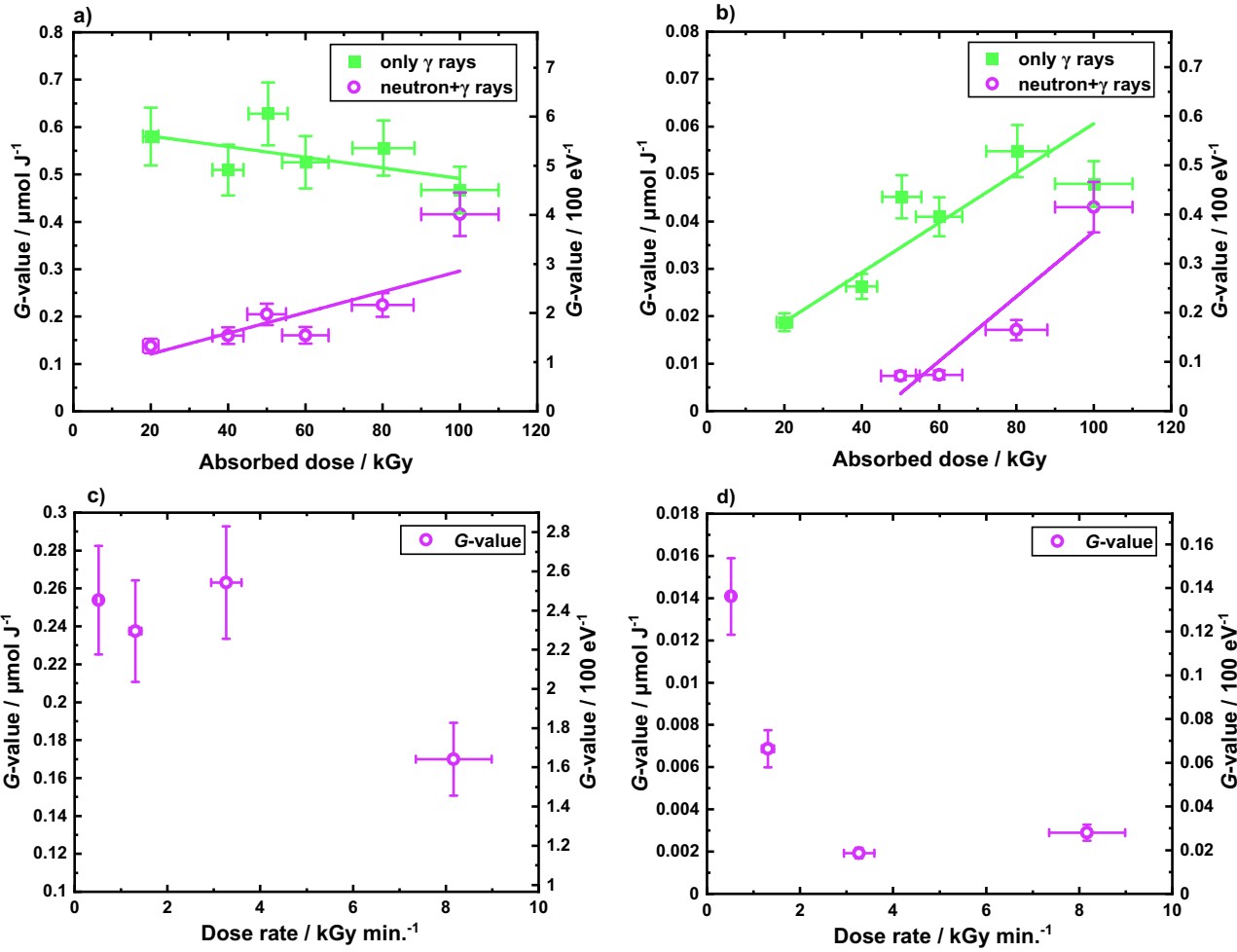

**Fig. 1 Radiation-chemical yields (*G*-values) of acetol and solketal from neat glycerol.** Given for the specified dose of either γ-ray (green squares) or neutron+γ-ray (magenta open circles) irradiations: (**a**) acetol and (**b**) solketal, as a function of absorbed dose, and (**c**) acetol and (**d**) solketal, as a function of dose rate. Samples in (**a**) and (**b**) were irradiated with dose rates ranging between 18 and 40 Gy min$^{-1}$ for γ rays and between 1600 and 6500 Gy min$^{-1}$ for the mixed field, while samples in (**c**) and (**d**) were irradiated with 50 kGy of mixed-field absorbed dose. The relationship between the two *G* quantities is 1 molecule (100 eV$^{-1}$) ≡ 0.1036 μmol J$^{-1}$. (Data and linear analysis are available in Supplementary Tables 2, 3 and 4); x-axis error bars derive from absorbed dose uncertainties, y-axis error bars represent the combination of errors from the relative standard deviation% (RSD%) of analyte concentration curves and absorbed dose uncertainties for each sample.

chemical yields of $486 \pm 160\,\mu g\,ml^{-1}$ and $0.17 \pm 0.06\,\mu mol\,J^{-1}$, respectively. These concentrations would produce a weakly acidic environment, with a calculated pH of 3.42. With an equivalent 50 kGy dose (Supplementary Table 6 and Supplementary Fig. 1), both modes of irradiation produce similar radiation-chemical yields of acetic acid within uncertainties.

The irradiations of most oxygen-containing organic compounds are unselective in terms of the variety of radiation-induced chemical components that are formed, i.e., ethanol, glycerol and acetone producing 18, 12 and 14 different stable chemical products[29–31], respectively. These observations suggest it is possible to augment the production of acetol and solketal using selected mixtures, absorbed doses and dose rates and that a process exists capable of yields >1 μmol J$^{-1}$, surpassing yields observed previously for either acetone or glycerol alone. Previously reported yields of any individual species from either acetone or glycerol did not exceed 0.27 μmol J$^{-1}$ for either diluted (<0.5 mol dm$^{-3}$) or neat samples[29,32,33]. This research expands the range to higher aqueous concentrations (>2 mol dm$^{-3}$) of glycerol and demonstrates higher yields for acetol (1.8 μmol J$^{-1}$) and consistent with dilution by water increasing acetol production over that for neat samples.

**Production mechanisms.** "Direct action" radiolysis mechanisms are often significant with pure or with solute concentrations above 10 wt.%[4,27]. This is in contrast to the "indirect" effects observed in dilute solutions (<10 wt.% solute) where the solvent (typically H$_2$O) takes the majority fraction of the absorbed energy, thus the accompanying chemical changes are indirectly initiated by radiolytic water species as reported in most of the associated literature. Glycerol has yet to be discussed specifically as to its direct effects from highly ionizing radiation. By extrapolating the known trends and mechanisms from similar alcohols[27,34], as well as the results presented in this paper, the direct effects on glycerol and the subsequent reaction kinetics can be postulated. Due to the numerous possible reactions, only relevant, probable reactions will be described.

Direct ionization and subsequent ion fragmentations on glycerol are shown in Fig. 3a, b, producing excited radical cations and non-ionizing electrons (C$_3$H$_8$O$_3$•$^+$, e$^-$). The electrons solvate after ~10$^{-10}$ s[35], the excited radical cations can react several ways via fragmentations and will contribute to the majority of C-O and C-C cleavages which neutralize, to result in the synthesis of the observed smaller compounds such as formaldehyde, acetaldehyde, methanol and acetone.

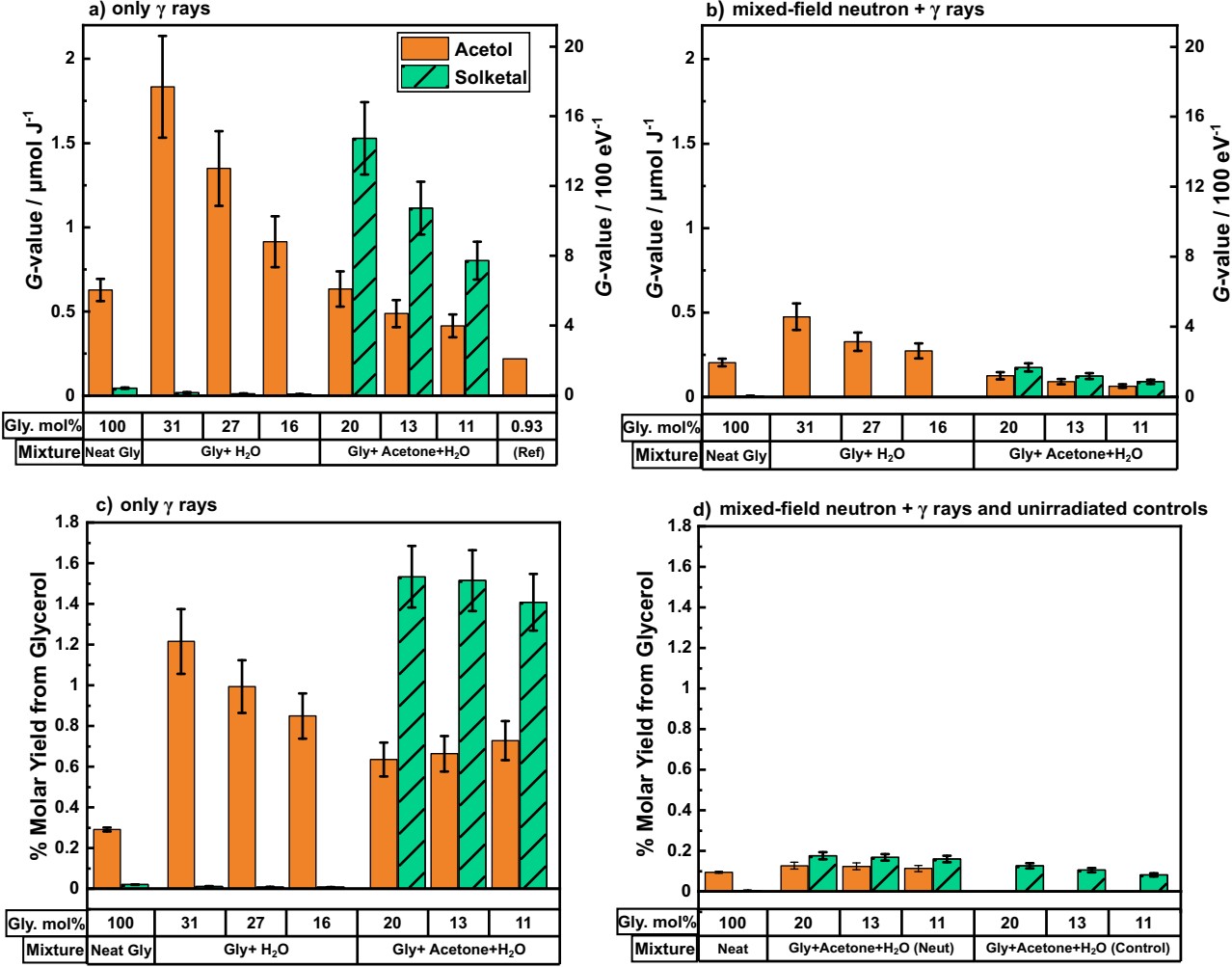

**Fig. 2 G-values and % molar yields for acetol and solketal from glycerol mixtures.** After 50 kGy irradiations of binary aqueous and ternary glycerol mixtures. **a** G-values for only γ-ray irradiations with an average dose rate of 40 Gy min⁻¹. **b** G-values for mixed-field neutron+γ-ray irradiations with a dose rate of 3260 Gy min⁻¹. **c** % molar yields for only γ-ray irradiations. **d** % molar yields of mixed-field and unirradiated control samples. The indicated ternary mixtures containing glycerol, acetone and water have the following compositions in mol.%: (i) 20, 20, 60; (ii) 13, 21, 65; and (iii) 11, 32, 56, respectively. (See Supplementary Table 5 for data.) G-value error bars represent the combination of errors from the RSD% of analyte concentration curves and absorbed dose uncertainties for each sample. %Molar yield errors derive only from the RSD% of analyte concentration curves for each sample. Reference data from Baugh et al.: (Ref)[29], for acetol used in (**a**), used $N_2O$-saturated aqueous glycerol samples with a γ-ray dose and dose rate of 1.4 kGy and 8 Gy min⁻¹, respectively.

Applying the Samuel–Magee theory[36–38], energetic volumes of reactive species termed *spurs* or *blobs* are formed after ionizing particle interactions within liquids. This theory allows the description of subsequent radiation-induced, diffusion-controlled ionic or radical reactions that occur in the surrounding solution of a spur site after ~$10^{-12}$ s from the ionization event. From water specifically, these reactive species can exist until ~$10^{-4}$ s after the initial event[39]. Initial intraspur ionic reactions from the ionized glycerol cation are given in Fig. 3c, d, producing an acidified glycerol cation which could be stabilized by the clustering mechanism given in Fig. 3e. Intraspur radical reactions typically result in C–H cleavages via H-abstraction mechanisms, forming either α-hydroxy (•C–O) or alkoxy (C–O•) radicals, as seen in Fig. 4a, b. Initially formed in comparable quantities inside the spurs, alkoxy radicals convert to the more stable carbon α-hydroxy radicals as the spur expands. For glycerol, radical-initiated H-abstraction from the secondary carbon is most probable which produces the more stable α-hydroxy radical as indicated Fig. 4c. This radical can be then converted to acetol through a previously suggested[29] radiation-triggered acid-

catalyzed water elimination rearrangement and subsequent radical chain-reaction propagation mechanism as expanded upon in Fig. 4d, e. For the direct action on glycerol for acetol production, the acid-catalyzed rearrangement mechanism is suggested here to be catalyzed by the stabilized, acidified glycerol cation ($CH_2OH–CHOH–CH_2OH_2^+$)[27,40] as generated within spurs via reactions in Fig. 3c–e.

Similar to other acidic solvent species (such as the hydronium ion, $H_3O^+$ from $H_2O$), the synthesis of this short-lived acidic catalytic species can be influenced by two irradiation factors which support the difference in yield and % molar yields for the two irradiations (Figs. 1, 2). The first factor relates to the difference in linear energy transfer (LET) between γ rays and neutrons manifest as a difference in radiation-chemical yield of the acidic species and consequently the rate of the acid-catalyzed mechanism for acetol formation.

For γ-ray (low LET) irradiations, low-energy interactions occur throughout the medium, creating small, well-separated volumes of energetic reactive species whereas the larger energetic volumes from energetic ions arising from neutron interactions can

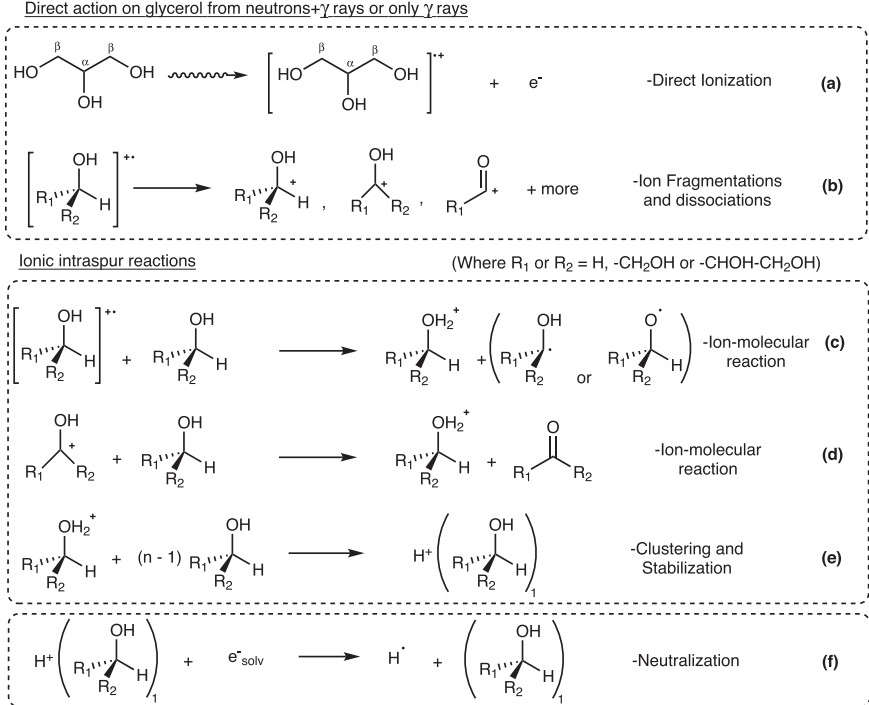

**Fig. 3 Proposed selected initial mechanisms from "direct action" upon pure neat glycerol. a** Direct ionization. **b** Ion fragmentations resulting in most C–C or C–O scissions. **c, d** Ion dissociation reactions forming acidic glycerol cations and with either radicals or carbonyl species. **e** Clustering and stabilization of acidic glycerol. **f** Neutralization of acidic glycerol via solvated electrons. Since H-dissociation could occur at any hydroxyl carbon of glycerol, $R_1$ and $R_2$ = –CH₂OH for α-carbon or –H and –CHOH–CH₂OH for β-carbon species, respectively.

overlap[41]. This overlap increases the probability of the interspur reactions which reduce the radiation-chemical yield of both the α-hydroxy radical and the stabilized glycerol cation as indicated in Supplementary Fig. 2a–e. When shutdown, residual radioactivity in the reactor emits γ rays that are low LET ($\approx$0.3 keV μm⁻¹)[39] whereas, when critical, the γ rays are joined by fission neutrons with LET from 10 to 100 keV μm⁻¹ [42]. This difference in radiation-chemical yield of an acidic solvent molecule (such as for the acidic glycerol cation) is evident for $H_3O^+$ from $H_2O$ (as shown in Fig. 4e) with contrasting LET irradiations, with the $G$-values for ⁶⁰Co γ rays (LET: 0.23 keV μm⁻¹) and α−particles (LET: 108 keV μm⁻¹) being 0.28 μmol J⁻¹ and 0.044 μmol J⁻¹, respectively[43].

A similar dependence on the short-lived acidic species is hypothesized here for ketalization reaction to form solketal, provided sufficient availability of acetone in the starting mixture, as indicated in Fig. 4g. The reaction proceeds chemically, with the necessary radical combinations being highly improbable. Acetone as a limiting reagent, explains the lower yields (a factor of 10 lower) of solketal compared with acetol in the neat samples. The increase in the $G$-value trend observed in Fig. 1b with absorbed dose is also explained by the accumulation of acetone for the solketal reaction for higher doses. Another proposition could assume that the ketalization reaction is mildly catalyzed by acetic acid[44]. Although both irradiations gave similar yields of acetic acid (shown in Supplementary Fig. 1), the data indicated in Fig. 2 indicate that this acid-catalysed process is heavily radiation-dependent and, more specifically, dose-rate dependent.

The second radiation factor relates to dose rate which is linked[43] to the yields of the reactive species such as the stabilized glycerol cation or $H_3O^+$ that participate in diffusion-controlled mechanisms. The higher dose rate observed for mixed fields compared to γ radiations would be similar to increasing LET[45], due to the increased probability of spurs overlapping and higher recombination rates of

the reactive species. Diffusion-controlled acetol and solketal mechanisms requiring such species would be limited which is supported by the data in Fig. 1c, d and Fig. 2. This is supported by the dose-rate dependence for solketal data in Fig. 1b.

As glycerol concentration is reduced via aqueous samples, the indirect effects upon glycerol become more important. After $\sim$10⁻¹² s, reactive species from direct water radiolysis (Fig. 4f) start to interact chemically with the solute, glycerol. Hydroxyl (HO•) and hydrogen (H•) radicals are reactive towards alcohols, typically initiating α-carbon H-abstractions shown by Fig. 4c. The hydronium ion, $(H_3O^+)$ is thought to also act as a catalyst for acetol and solketal production in diluted solutions, like the acidified glycerol cation in neat samples. Lastly, the solvated electron ($e_{solv}^-$) being fairly unreactive towards alcohols is however fairly active towards carbonyls groups[27] such as in acetol. Therefore, the decrease of γ-ray acetol $G$-values with increased dose as indicated in Fig. 1a is explained due to the conflicting reaction in Supplementary Fig. 2f. For the ternary mixtures in Fig. 2, with acetone no longer a limiting reagent, there would competition for the short-lived acidic species between the acetol and solketal processes shown in Fig. 4, reducing acetol $G$-values compared to binary mixtures.

For the diffusion-controlled reactions of acetol and solketal, the reaction rates, $k_D$ depends on the diffusion constants, $D$ of the required respective species shown in Fig. 4. The diffusion constants are influenced by two related factors[46]; (a) viscosity, $\eta$, and (b) temperature, $T$ as indicated by the Stokes–Einstein relationship, Eq. (1):

$$k_D \propto D = \frac{kT}{6\pi\eta R} \qquad (1)$$

where $k$ is Boltzmann's constant and $R$ is the species' radii. This dependence can help explain the trends observed between Figs. 1, 2 for acetol and solketal production. The dilution of glycerol will

**Fig. 4 Subsequent radiation-directed, diffusion-controlled reactions.** Radical-directed reactions (**a–c**): (**a**) Intraspur radical conversion to the more stable hydroxy radical. **b** C–H scission dominated by α-hydrogen abstraction. Reactions (**c–e**) is the expanded mechanism for acetol production[29]. **c** Hydrogen abstraction at the weaker α-C–H bond. **d** Acid-catalyzed chain rearrangement via an acidified species. **e** α-H-abstraction and α-hydroxy radical regeneration. **f** Direct action on water producing reactive species such as $H_3O^+$. **g** Acid-catalyzed chemical ketalization reaction scheme for solketal ($R_1$ and $R_2 = -CH_2OH$ for α-carbon or H and $-CHOH-CH_2OH$ for β-carbon radicals, respectively).

decrease the viscosity of the solution[47], improving diffusion and reaction rates for Fig. 4 reactions.

The additional heating effect caused by increased absorbed doses would raise the temperature of the solution, lower viscosities, increase diffusion constants and increase the reaction rates for diffusion-controlled reactions, such as those for acetol and solketal. For water radiolysis, there is an estimated 24 °C temperature increase for 100 kGy absorbed dose[27], compared with a 4.8 °C rise at 20 kGy. Thermal energy would be able to dissipate for longer γ-only irradiations, whereas for higher dose rate mixed-field irradiations, the thermal energy would accumulate. These two sample factors explain; (i) the increase in *G*-values seen from neat to diluted glycerol samples, and (ii) increasing acetol *G*-values with absorbed dose of mixed-field neutron+γ-ray irradiations (Fig. 1a).

The short-lived acidic catalytic species ($R_1R_2CHOH_2^+$ or $H_3O^+$) which are required for both acid-catalyzed reactions are neutralized by solvated electrons ($e_{solv}^-$), as shown in conflicting reactions of Fig. 3f. and Supplementary Fig. 2g. Additional conflicting reactions[4,27] involving molecular oxygen ($O_2$) are presented in Supplementary Fig. 2h–k, $O_2$ acts as a scavenger for H• and α-hydroxy radicals, reducing their concentrations and inhibiting acetol synthesis.

The scavenging of $e_{solv}^-$ for future research is proposed to improve the *G*-values, production yields and radiolytic process viability of both acetol and solketal. Additionally, the complete removal of $O_2$ from the samples to promote concentrations of required radicals would also increase acetol production. It is postulated that *G*-values of $> \sim 20 \, \mu mol \, J^{-1}$ for the chain reaction could be achieved, as observed in similar polyols[48]. Processes of freeze-drying and $N_2O$ saturation would be necessary and may increase production costs.

**Process analysis: modelling scaling-up scenarios.** To examine the potential to scale up this current process and hence its industrial feasibility, particle transport simulations have been performed with the Monte Carlo particle transport code (MCNP)[49] to determine γ-ray dose rates. These have then been used to predict the maximum annual production of solketal and acetol for the two glycerol mixtures with the highest observed *G*-values (Fig. 5a) from data obtained using the TRIGA Mark II reactor (Fig. 5b) and then extrapolating for two hypothetical industrial-scale scenarios:

  i. A 2 GWth, 2-loop Pressurized Water Reactor (PWR), with production in glycerol irradiated in pipes routed in the

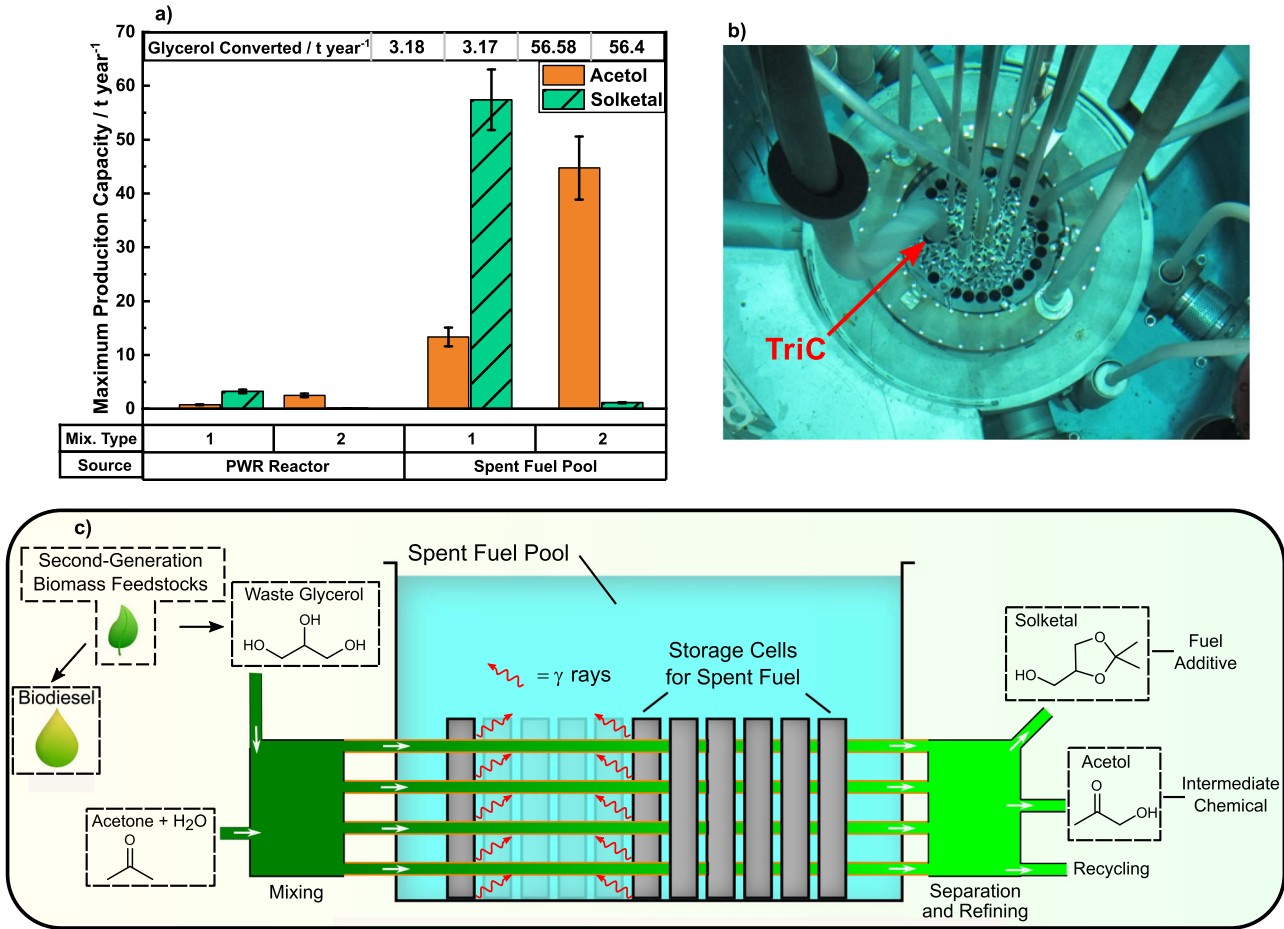

**Fig. 5 Annual production capacity and scale-up process models. a** Solketal and acetol annual production capacity for (1) ternary acetone, glycerol and water mixture with mol% of 20, 20, and 60, respectively, and (2) a binary glycerol-water mixture with 31 mol% of glycerol. Error bars indicate the propagation of errors from the empirical *G*-value data. The empirical data (Fig. 2) was obtained using (**b**) the 250 kW TRIGA reactor and the shown Triangular Irradiation Channel (TriC) for irradiations. For scale-up purposes, the two irradiation scenarios were based on geometrical models in MCNP, as per Supplementary Fig. 3. **c** Nuclear-biorefinery process schematic depicting a spent fuel storage pool; an equivalent scenario is also plausible for the case of dry storage.

| Table 1 Impact values of solketal production on transport fuels. | | |
|---|---|---|
| **Impact values** | **(i) PWR spent fuel pool (One NPP)** | **(ii) Max. capacity within Europe (180 NPPs)** |
| Annual solketal production (t year$^{-1}$) | 57 ± 6 | $(1 ± 0.1) × 10^4$ |
| Annual solketal production (litres year$^{-1}$) | $(5.4 ± 0.5) × 10^4$ | $(9.6 ± 1) × 10^6$ |
| Total annual petrol blend volume *5% solketal, 95% vol% base petroleum* (litres year$^{-1}$) | $(1.1 ± 0.1) × 10^6$ | $(1.9 ± 0.2) × 10^8$ |
| (i) PWR spent fuel pool and (ii) 180 equivalent spent fuel pools within Europe in 2021. | | |

cavity between the reactor pressure vessel and the concrete, biological shield (Supplementary Fig. 3a).

ii.  A vessel containing 10 spent nuclear fuel elements with dose rates determined in a pipe at the middle of the vessel, extrapolated to 1780 total elements for capacity calculations (Supplementary Fig. 3b).

These production values have been calculated by combining empirical *G*-values with dose and dose rate values obtained from MCNP models. The TRIGA model resulted in negligible values for annual production capacity relative to the PWR and spent fuel models.

This analysis suggests that the PWR spent fuel pool, utilizing only γ rays, has the greatest hypothetical scale-up production capacity of 57 ± 6 t year$^{-1}$ and 13 ± 2 t year$^{-1}$ of solketal and acetol, respectively, whilst consuming 57 ± 8 t year$^{-1}$ and 25 ± 2 t year$^{-1}$ of glycerol and acetone, respectively. The main factor responsible for the optimal production capacity is the advantageous volume available for irradiation in this case. By way of another potential advantage, γ-only irradiations of a spent fuel pool remove the potential for neutron activation of metal impurities, especially when high-sodium, crude glycerol could be considered as a feedstock. Table 1 lists values to show the potential impact of a radiation-chemical plant and its expansion

to the available spent fuel sites within the geographical area of Europe[50] (shown by country in Supplementary Fig. 4).

The maximum solketal production capacity from the hypothetical use of spent fuel facilities in Europe considered in this work is $\sim 1 \pm 0.1 \times 10^4$ t year$^{-1}$. This would alleviate forecast demand for biomass-derived, sustainable fuel additives which are expensive. Notwithstanding the scarcely available data[51] for solketal prices ($\approx 3000$ \$ tonne$^{-1}$), it could be produced via irradiations at lower costs than suggested chemoselective methods ($\approx 2088$ \$ tonne$^{-1}$). This would be due to negligible radiation processing costs[27] from spent fuel sources if process yields and selectivity can be improved to match chemoselective methods. However, solketal production costs would depend on feedstock prices, a nuclear-biorefinery plant concept would avoid feedstock purchasing market issues and costs since both biodiesel and solketal could be produced on the same site. Given the renewable proportion added to reduce the volume of petrochemically-derived fuels will increase from 5% in current blends to 20%[52] by 2030[53], solketal sourced this way could offer a renewable alternative whilst consuming waste organics, addressing net-zero targets[54,55], improving the value proposition of nuclear energy and pioneering a beneficial use of radioactive waste.

**Immediate suitability of the scheme for integration with nuclear facilities**. NPP design, commissioning and operation is a highly regulated activity that frequently takes decades to complete, and this is often only in the most efficient of projects; this can be an important factor associated with the accurate estimation of upfront costs described earlier and which motivated this research. However, the merit of the process described in this research is that in particular, we highlight the benefit of the γ-only scheme, not only because it avoids problems associated with neutron activation products pervading the process infrastructure associated with the organic feedstock production, but also because it removes the need for this scheme to mandate integration with an *operating* reactor. This removes the need for the neutronic design demands of a reactor to be augmented to accommodate the feedstock production process, and it also removes the less desirable combination of risks associated with organics production and the need to maintain reactor stability during operation. Even if achievable, the scope for the additional source of operational interruptions due to the organic side of the process aspect is undesirable, since a further key factor in the economic viability of nuclear power is the need for the high duty factors associated with long-term, uninterrupted operation.

Concerning the γ-only scheme, the nuclear industry has a long history of operating processes at a commercial scale that manage the hazards associated with significant quantities of fissile material, flammable organophosphorus compounds (i.e., tri-*n*-butyl phosphate), highly corrosive conditions and high levels of radiation. This has been done principally for the purpose of reprocessing spent nuclear fuel, with such operations achieved with an extraordinarily high degree of regulatory compliance over many decades. By comparison, the hazard potential associated with the integration of the organic production scheme described in this research at scale, utilizing γ radiation from spent nuclear fuel in interim storage, could be reduced markedly because the organic media and the spent fuel would be separated physically from each other, with the potential for inadvertent nuclear criticality eliminated by the design of spent fuel storage facilities (rather than being minimized by safe geometry vessels as is the case in reprocessing), whilst no extreme temperatures or pH are necessary for the process to be enacted. Further, the recovery of the valuable feedstocks could be achieved at a site remote from the irradiation facility. This possibility notwithstanding, for future nuclear-chemical co-generation processes to be successfully enacted, research into process modelling, corrosion, and safety aspects would need to be completed. However, it is worthy of note that the world has significant stockpiles of spent nuclear fuel which, which constitute a significant intergenerational societal issue, have to be stored safely whilst serving no long-term useful purpose and, in many cases, neither is a long-term disposal strategy is in place for this material. The prospect we describe brings value and focus to this otherwise unending prospect whilst appealing to the climate imperative for there to be renewable alternatives to organic feedstock production.

## Conclusions

In summary, we have developed a radiation-chemical process for acetol and solketal production from the renewable feedstock, glycerol. Utilizing a 250 kW research fission reactor, we report solketal G-values of $1.5 \pm 0.2$ μmol J$^{-1}$, and improved acetol G-values of $1.8 \pm 0.2$ μmol J$^{-1}$ using γ-ray absorbed doses of >20 kGy, high concentration (>11 mol%) glycerol samples and ternary glycerol–acetone mixtures. A mechanistic discussion on the direct action upon glycerol has been explored with several important species identified such as the acidified glycerol cation, hydroxy radicals and $H_3O^+$ involved in acetol and solketal production. Using spatial MCNP models, the empirical results were expanded for a theoretical nuclear co-generation system involving; (i) a 2 GW PWR and (ii) a spent fuel pool with 1780 elements. These models show that the greater radiation volume of the spent fuel pool is advantageous for superior annual production totals. Further expansion to a potential 180 spent fuel sites within Europe revealed a maximum production capacity of $\sim 1 \times 10^4$ $(\pm 1 \times 10^3)$ t year$^{-1}$ for solketal. While this radiation process may not compete currently with other chemoselective pathways, it constitutes the first-reported, radiation-induced chemical process for solketal, and provides a notable example of the potential for unexplored renewable processes that can be realized using ionizing radiation - especially considering waste spent fuel pools as a source of catalytic energy.

## Methods

**Materials and sample preparation**. For the irradiation mixtures, glycerol (>99.5 mass %) was purchased from Honeywell while acetone (99.8 mass %) was purchased from Fisher Scientific. Ultra-pure water was used from a Milli-Q Direct purification system. Chemical analytical standards for acetol (95 mass %), acetic acid (99.9 mass %), 1-butanol (99.9 mass %) were purchased from Sigma Aldrich. Ethanol and propanol used for the pre-chemical analysis of samples were also purchased from Sigma Aldrich. Solketal (98 mass%) was purchased from Alfar Aesar. Chemicals were used without further treatment. All liquid mixtures were prepared gravimetrically using a Fisherbrand FB73651 analytical balance with a stated accuracy (repeatability) of ±0.1 mg. The same balance was used in post-irradiation sample dilutions, the mass measurement errors can be considered negligible when compared against the calibration curves and absorbed dose calculation errors. Polypropylene Argos cryovials of 5 mL were purchased from Fisher Scientific and used as irradiation vessels.

**Irradiations**. Organic samples were irradiated using a TRIGA Mark. II fission research reactor at the Jožef Stefan Institute (JSI), previously described in the literature[56]. This light water reactor uses fuel elements made of 20% enriched $^{235}U$ within a zirconium hydride composite. It has a maximum steady-state power of 250 kW and has a maximum neutron and γ fluence of $1.9 \times 10^{13}$ cm$^{-2}$ s$^{-1}$ and $2.1 \times 10^{13}$ cm$^{-2}$ s$^{-1}$, respectively, within its central irradiation channel (CC)[57]. All the organic samples were irradiated in the larger triangular irradiation channel (TriC) and irradiated with either: delayed γ (only γ-rays) when the reactor was shut down, or a mixture of neutrons and prompt γ (neutrons+γ) when the reactor was critical.

For the study of the absorbed dose dependence, samples were irradiated with doses of either 20, 40, 50, 60, 80 or 100 kGy for each reactor mode/irradiation type. Dose rates ranged from 15.8 Gy min$^{-1}$ to 40.5 Gy min$^{-1}$ for γ-irradiated samples and 1630 Gy min$^{-1}$ to 6540 Gy min$^{-1}$ for neutron-irradiated samples. For the study of dose rate dependence, neat glycerol samples were irradiated with 50 kGy at different reactor powers of 16, 40, 100 and 250 kW, with dose rates of 520, 1310, 3270 and 8170 kGy min$^{-1}$, respectively. For comparisons between mixture types, all new samples of each respective radiation mode were irradiated during the same

run for 50 kGy. The samples exposed to γ were irradiated with 40 Gy min$^{-1}$. The samples exposed to the mixed neutron+γ field were irradiated with a dose rate of 3260 Gy min$^{-1}$. Post-irradiation, all samples were placed within a freezer at −20 °C until transport to Lancaster University (UK) for chemical analysis. All control samples are transported, stored in and analyzed using the same procedures alongside the irradiated samples.

For γ irradiations during reactor shutdown, it is measured that the γ dose rate in Gy s$^{-1}$ is proportional to the power reading on reactor instrumentation in watts (i.e. linear channel, which is a compensated ionization chamber and is sensitive also to the delayed γ rays—the result is presented in the units of W). Using a calibrated ionization chamber, a factor of 14,250 Gy s$^{-1}$ W$^{-1}$ for triangular irradiation channel was determined with the accuracy of 10% and was utilized for γ-dose determination. Due to the logarithmic decay of the reactor power (<1.5 kW) and consequently changing dose rate, the average power was taken for short time intervals and the absorbed dose totalled over time until the required dose was achieved.

For mixed-field irradiations during reactor operation, absorbed dose determinations were supported by calculations using the calibrated, validated JSI TRIGA reactor MCNP model[28,58,59] in addition to determining dose uncertainties. When reactor power is high enough (>10 kW), the delayed γ rays represent approximately 20% of the total dose.

**Chemical analysis.** All irradiated samples were analyzed within 30 days of their irradiation and 40 days of preparation. All samples were volumetrically diluted with ethanol in a ≈15:1 mass ratio with gravimetric measurements conducted throughout due to glycerol's high viscosity. Calculated average densities were and utilized for volumetric dilutions of glycerol mixtures. 40 μl of a 1 mg ml$^{-1}$ diluted stock solution of the internal standard, butan-2-ol was added to each sample for the internal standard calibration methodology. Samples were analyzed using a Shimadzu TQ8040 gas chromatography-mass spectrometer (GC-MS) equipped with an AOC 6000 autosampler. Shimadzu's LabSolutions GC-MS software was utilized for data capture, analyte confirmation using analytical standards, and further quantitation analysis. The same software was used as an interface for comparisons between the measured fragmentation patterns and the NIST 11 MS standard reference database. The separations were performed using a 10 m column guard and a Zebron 624-Plus analytical column; with a length of 30 m × 0.25 mm i.d. and a film thickness of 1.4 μm. The injector temperature was set to 300 °C and the oven program was set as follows: 40 °C (10 min); ramp of 25 °C min$^{-1}$ to hold at 300 °C (2.6 min). Split injections were used with a volume of 1 μl, with a split ratio of 20:1 with a constant column flow of 1.71 ml min$^{-1}$ during a run. The carrier gas used was helium with a purity of 99.999%. The detector and interface temperatures were set to 250 °C and 300 °C, respectively. For the MS detector, it was set to full scan mode at a scan speed of 1000 da s$^{-1}$ between the mass-charge ratio (m/z) range of 30–300.

The concentration of the radiolysis products within the diluted samples was directly measured through the use of internal calibration curves. Total product moles were calculated from the concentration by adjusting for the mass fragment extracted and the volumetric dilution. The radiation-chemical yield values (G-values) were calculated using the total analyte moles and the energy into the organic sample which was calculated using the absorbed dose calculations and the starting mass of the organic sample. Errors for the concentrations were derived from the relative standard deviation (RSD%) of the specific calibration curve used. The final error calculations for the radiation-chemical yields were determined using RSD% of the initial analyte concentration, the uncertainty in volumetric and gravimetric dilutions, as well as the uncertainty for absorbed dose. Uncertainty for G-value data points is in a range of ±(11–15%) depending on the sample.

**Particle transport simulations.** Particle transport simulations were performed to determine gamma dose rates. The simulations were performed using MCNP (Monte Carlo N-Particle) transport code[49]. MCNP has been validated on several benchmark experiments in the field of reactor physics, radiation shielding, particle accelerators, medical applications, etc. For this work MCNP 6.1.1. was used on one node of a modern computer cluster with 40 cores/80 threads (Intel® Xeon® Gold 6148).

The JSI TRIGA Mark. II MCNP model has been validated using several experiments[58,60,61] and has proven to be accurate in determining γ and neutron dose fields. An irradiation sample corresponding to the experimental setup was added to the model and γ fluxes were tallied inside of the container which was filled with glycerol. For this analysis, the γ H*(10) ambient dose equivalent was calculated using flux-to-dose conversion factors from the ICRP-21 report[59] and using the JEFF-3.3 nuclear data library[62] for all three cases described in this chapter.

The typical PWR MCNP model was developed at JSI for determining dose fields throughout the containment building and determining the detector responses in the biological shield surrounding the reactor pressure vessel. Tubes (4 m height, 4.8 cm inner radius, 5 cm outer radius) made of stainless steel coated in a layer of indium (2 mm, produces additional γ rays via the neutron capture reaction) and filled with glycerol were added to the model. The tubes were positioned in the reactor cavity between the pressure vessel and the biological shield. The cavity could potentially accommodate 50 tubes. The simulations were performed for the case of an operating reactor resulting in a mixed γ and neutron field. Because of the

large attenuation between the particle source and the pipes where the doses need to be calculated, variance reduction of the particle transport simulation was needed. The ADVANTG code[63] was used to prepare effective variance reduction parameters.

The spent nuclear fuel pool model was constructed for this analysis. Ten fuel elements from the typical PWR model were modelled in a tank of borated water. The γ particle source spectrum and activity were determined based on a typical burnup scenario (46274.21 MWd/tU). It should be noted that only one stainless steel pipe (2 m long, 4.8 cm inner radius, 5 cm outer radius) filled with glycerol at the middle height of the fuel elements (at 183 cm) was modelled.

**Scale-up calculations.** To determine the maximum yearly production capacity of each facility/model, radiation-chemical yields and starting mixture characteristics given in Fig. 2 were combined with physical values determined from MCNP models. These values include delayed γ dose rates which are comparable to the empirical data obtained using the TRIGA reactor. The PWR reactor MCNP model was expanded for fifty pipes within the walls of the reactor vessel, with a total organic irradiation volume of $2.9 \times 10^4$ m$^3$. The $2 \times 5$ spent fuel pool matrix MCNP model which carries the organic mixture was extended for ten 0.1 m × 12 m pipes in the vertical axis. The volume for irradiation was then expanded to the maximum operational capacity of 1780 elements for the PWR spent pool, totalling 560 mixture-carrying pipes with a total irradiation volume of $9.4 \times 10^5$ m$^3$. Mixtures yielding the highest G-value for solketal and acetol were explored for both models. For consistency with the empirical data, scaled-up volumes would be irradiated with 50 kGy of absorbed dose. For the yearly maximum production capacity of solketal within the geographical area of Europe, it is assumed that the 180 NPPs would have similar spent fuel facilities as the typical 2 GW PWR facility.

## Data availability
The data generated during this study are included in the published article and the Supplementary Information, Table 1–6.

## Code availability
The MCNP code for particle transport is available via the RSICC (Radiation Safety Information Computational Center). The input for the MCNP code is available from the corresponding authors upon a reasonable request.

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

## Acknowledgements

The authors would like to thank Dr David Rochester for his technical assistance in the chemical analysis stages. The authors wish to acknowledge Lancaster University, the Engineering and the Physical Sciences Research Council (EPSRC) for supporting the Ph.D. studentship of A.G.P.

The authors (B.K., L.S.) acknowledge the financial support from the Slovenian Research Agency (research core funding No. (P2-0073). The authors (A.J., L.S.) acknowledge the financial support from the Slovenian Research Agency (research infrastructure core funding TRIGA Mark II research reactor). M.J.J. acknowledges the support of the Royal Society via a Wolfson Research Merit Award.

## Author contributions

V.N.V. and M.J.J. contributed equally to the original premise for the research, interpretation of results, coordination of the paper, and jointly supervised the related Ph.D. studies of A.G.P. M.J.J. contributed the "immediate suitability of the scheme for nuclear facilities" section. A.G.P. was the primary author of the paper, led the empirical experimental design, chemical analysis, scale-up calculations and the interpretation of results. B.K. performed the simulations of particle transport in support of the measurements. He is also the author of the typical PWR and spent nuclear fuel pool MCNP models used for the theoretical scaling up of the production. A.J. performed irradiation of samples and dose determination. L.S. contributed to interpretation of results, author of

the TRIGA MCNP model, modelling of dose rates, original premise for the transition from research reactor to NPP reactor and spent fuel pool systems.

## Competing interests

The authors declare no competing interests.
