## [Transparent Peer Review File · Communications Chemistry]

Reviewers' comments:

Reviewer #1 (Remarks to the Author):

This manuscript discusses nuclear-driven production of renewable fuel additives from waste organics with particular focus on solketal and acetol production from glycerol-acetone mixtures. The use of ionizing radiation from an operating nuclear reactor or from the used nuclear fuel stored after removal from the reactor core in driving processes of valuable feedstocks is indeed very interesting. However, a discussion about this should also include the numerous safety aspects that must be considered. Also, the more scientific part of the manuscript, i.e., the measured G-values and the corresponding mechanistic discussion would need to be deepened before publication is considered. More specifically, I would encourage the authors to act upon the following comments:

1. Page 2. The statement "The energy converted into electricity in a reactor has a value of 10^{-3} to 10^2 \$ mol⁻¹, whereas ionizing radiation produced by a reactor can yield chemicals via radiolysis of value 10^3 to 10^6 \$ mol⁻¹" requires some clarification. I am not sure that it is obvious to the general reader what "mol⁻¹" refers to in this case (it is at least not obvious to me). Value per mol of what?
2. The authors make use of the term "radiochemical yield" throughout the manuscript. For a radiochemist this term has a completely different meaning. I would suggest that the authors use the term "radiation chemical yield" which makes sense both to radiation chemists and radiochemists.
3. The radiation chemical yields (G-values) presented in Figures 1 and 2 constitute the core of the scientific results presented in this manuscript. However, there is no detailed mechanistic discussion related to the trends that can be observed in these figures. The only reactions that are presented are those in Figure 3 and they can hardly be considered as complete. To make this manuscript sufficiently interesting from a chemical point of view, a much more thorough discussion on the mechanism in general and in connection to the results presented in Figures 1 and 2 must be added. The following points highlight some observations that must be included.
4. Figure 1 a and b present G-values that change with absorbed dose. Normally, G-values would describe the yields at very low conversions but here I assume that the G-value describes the total change at a given dose. Please discuss why the presented G-values are changing with dose from a mechanistic point of view.
5. The G-values for solketal are one order of magnitude lower than the G-values for acetol under the same conditions (Figure 1). Please provide an explanation to this (which I assume can be found in a more detailed mechanism).
6. Why do the G-values depend on dose rate?
7. Figure 2 shows some very interesting results for mixtures of glycerol and water and for mixtures of glycerol, water and acetone. It is clear that for the glycerol/water mixture (gamma irradiated), the yield of acetol at a given dose decreases with decreasing glycerol concentration. In the glycerol/water/acetone mixture, this is not the case. In this case, the yield of solketal is also very high (and independent of glycerol concentration). These observations should be more deeply discussed as they most certainly can prove mechanistic insights. Why is the yield of acetol lower in the ternary mixtures than in the binary mixtures? Competition kinetics? Is acetol converted into solketal?
8. When reading the manuscript, it is easy to get the impression that the formation of acetol and solketal can be attributed to acidification induced by radiation. If this would be the case, a reference experiment in buffered solution should be conducted in order to demonstrate this. Considering the very high concentrations of glycerol, I am surprised to see that direct radiolysis of glycerol is not discussed. Considering the high reactivity of both hydroxyl radicals and hydrogen atoms towards glycerol and acetone, I am also very surprised not to see any discussion on these possible reactions. These points must be addressed.
9. As mentioned above, the safety aspects of the proposed nuclear-driven production of renewable fuel additives are not mentioned. In general, introducing even the slightest modification in a nuclear facility requires thorough safety analysis. The proposed facility making use of the nuclear reactor itself or a storage pool for spent nuclear fuel would involve a major change in the nuclear facility with risks that are quite obvious in terms of risk for corrosion, leakage, contamination, fire, etc. In addition, it is

not clear how the integration of the fuel additive production facility and the nuclear power plant will affect the performance of the nuclear power plant. Since safety is such a crucial part of nuclear technology, I would strongly recommend the authors to include a paragraph addressing the safety issues.

Reviewer #2 (Remarks to the Author):

The manuscript devoted to the combination of technologies using ionizing radiation for the deposition of valuable raw materials from spent glycerin will certainly be of interest to specialists in these fields of science.

Reducing the amount of greenhouse emissions using the energy of the "peaceful atom" is a very promising direction. The use of a scientific reactor showed the consistency of the considered process. My recommendation is to determine the composition of unknown 46.5% of the components obtained in the process of radiolysis to assess their safety at various dose rates, since a wide variety of products can be formed, including those that pose a danger to personnel. In subsequent studies, I would like to see the thermodynamic characteristics of the process under conditions of exposure to ionizing radiation, as well as the analysis of gaseous products of radiolysis, which may call into question the use of this method for processing glycerin in the conditions of power reactors of nuclear power plants. Such initiatives, without proof to the contrary, can lead to emergency modes in the glycerin processing loop, which may affect the stability of the NPP operation. The world is aware of enough sad incidents at the enterprises for the reprocessing of spent nuclear fuel. Given the updated data, the design of the reactor may need to be modernized.

Have you performed dosimetry of the irradiated samples?

Surprisingly, the presentation of the material is not standard, there are no conclusions. There is no clear Introduction section ("must begin with the heading "Introduction""), no clear Results/Conclusions section ("must begin with the heading "Results""). Also, there is only one Table in the manuscript text, but it is named "Table 2" (should be Table 1 / Table).

As a result, the article can be accepted for publication with minor revision.

Reviewer #3 (Remarks to the Author):

The manuscript develops the argument that use of radiation from nuclear reactors to convert mixtures of glycerol and acetone to higher value products including acetol and solketal might improve the economics of nuclear power.

The paper presents radiochemical data with G values from irradiations of glycerol +/- water/acetone which appear to be carefully obtained with mechanistically sensible explanations.

Although addressing the cost of nuclear power and improving the economic value proposition is extremely important, and the use of otherwise un-utilized radiation from nuclear reactions might help in improving the economics, how to make use of radiation remains an open question. The nuclear radiation must replace something expensive in an existing chemical process in order to offer some cost advantage. Unfortunately, the authors have selected chemical processes that can already be performed under relatively mild conditions and the radiation is used to replace the low cost acidic catalysts already used, which offers little savings.

It will almost never make sense to make chemical products that are done in low capital low energy existing processes, it would be highly recommended that reactions with large energetic barriers or where expensive non-selective pathways are presently used be examined to provide clear incentives for use of nuclear technology.

Issues requiring attention from the authors:

1) The authors indicate glycerol as having zero cost, in fact, although early in the history of biodiesel the market was saturated and the price very low, global demand increased with new uses for glycerol and demand has outpaced supply. Prices now are approximately \$200/ton for crude glycerol and higher when refined. The authors must address the impact of feed cost on their conclusions. If radiochemistry is to be used it will be important to know the impact of impurities typical from the unrefined crude feedstock.

2) Reference should be made to the many papers with thermochemical pathways and simple acid catalysts to produce acetol from glycerol. Also, the technoeconomic analysis by Al-Saadi and colleagues for acid catalysed production of Solketal from acetone + glycerol, *Front Chem.* 2019; 7: 882. doi: 10.3389/fchem.2019.00882

The value of solketal ~ \$3/kg should be critically questioned.

3) The data presented indicates that gamma radiation is the most effective for the product selectivity and G's. This is also true because neutron irradiation of feeds with lots of impurities might activate impurities rendering the products radioactive and valueless. This point should be made very clear in the paper.

4) Although, it is always nice to see Monte Carlo radiation transport models, the value for this paper is unclear. Moving most all of that material to the supplementary information would make the major messages more clear.

5) Throughout the paper editing is needed – I refer that chore to the editors.

Response to Reviewers' Comments

Communications Chemistry – Nature Paper ID: COMMSCHEM-21-0127-T

Text in red – Reviewers' comments

Text in black – Authors' response

Reviewer #1 (Remarks to the Author):

This manuscript discusses nuclear-driven production of renewable fuel additives from waste organics with particular focus on solketal and acetol production from glycerol-acetone mixtures. The use of ionizing radiation from an operating nuclear reactor or from the used nuclear fuel stored after removal from the reactor core in driving processes of valuable feedstocks is indeed very interesting. However, a discussion about this should also include the numerous safety aspects that must be considered. Also, the more scientific part of the manuscript, i.e., the measured G-values and the corresponding mechanistic discussion would need to be deepened before publication is considered. More specifically, I would encourage the authors to act upon the following comments:

1. Page 2. The statement “The energy converted into electricity in a reactor has a value of 10^{-3} to 10^2 \$ mol⁻¹, whereas ionizing radiation produced by a reactor can yield chemicals via radiolysis of value 10^3 to 10^6 \$ mol⁻¹” requires some clarification. I am not sure that it is obvious to the general reader what “mol⁻¹” refers to in this case (it is at least not obvious to me). Value per mol of what?

We agree that these units are not clear. We have used the data published in ref. 2 and it is not clear how these were calculated. We were not able to clarify these data with the authors and, therefore, we have decided to delete the statement and substitute it with the following (Page 2):

“According to the comprehensive techno-economic analysis published by Schmeda-Lopez², integration of a large nuclear power plant and a chemical process using the reactor's γ radiation to facilitate the production of commodity chemicals such as propylene leads to significant economic benefits. This suggests an economically attractive route might exist to valorise waste biomass that avoids petrochemical production. “

2. The authors make use of the term “radiochemical yield” throughout the manuscript. For a radiochemist this term has a completely different meaning. I would suggest that the authors use the term “radiation chemical yield” which makes sense both to radiation chemists and radiochemists.

As suggested, the references to “radiochemical yield” have been changed to “radiation chemical yield” throughout the text to save any confusion to readers from different fields.

3. The radiation chemical yields (G-values) presented in Figures 1 and 2 constitute the core of the scientific results presented in this manuscript. However, there is no detailed mechanistic discussion related to the trends that can be observed in these figures. The only reactions that are presented are those in Figure 3 and they can hardly be considered as complete. To make this manuscript sufficiently interesting from a chemical point of view, a much more thorough discussion on the mechanism in general and in connection to the results presented in Figures 1 and 2 must be added. The following points highlight some observations that must be included.

The reviewer raises an important matter which we have considered in some detail. Significant alterations have been made to the discussion section to describe the direct action mechanisms upon glycerol as it is not discussed in the currently available literature. A newly added text is highlighted in yellows and spans from pages 7 to 12.

We propose direct action reactions upon glycerol given in Fig. 3, with additional spur reactions in Fig. 4. Conflicting side reactions have been listed in Fig. S2 of the supplementary material. The differentiation between direct and indirect effects has been included to clarify the different kinetics and species for neat and dilute samples, respectively.

Multiple references have been included to support these additional conclusions, namely:

Ref 27: Spinks, J. W. T. & Woods, R. J. in *An introduction to radiation chemistry* (1990).

Ref 34: Freeman, G. R. in *The Study of Fast Processes and Transient Species by Electron Pulse Radiolysis* 399-416 (Springer, 1982).

Ref 35: Muroya, Y. *et al.* Ultra-fast pulse radiolysis: A review of the recent system progress and its application to study on initial yields and solvation processes of solvated electrons in various kinds of alcohols. *Radiation Physics and Chemistry* **77**, 1176-1182, doi:<https://doi.org/10.1016/j.radphyschem.2008.05.035> (2008).

Ref 40: Grimsrud, E. P. & Kebarle, P. Gas phase ion equilibriums studies of the hydrogen ion by methanol, dimethyl ether, and water. Effect of hydrogen bonding. *Journal of the American Chemical Society* **95**, 7939-7943, doi:<https://doi.org/10.1021/ja00805a002> (1973).

Ref 46: Meisel, D. *et al.* Radiation chemistry of synthetic waste. (Argonne National Lab., IL (United States), 1991).

Ref 47: Wagner, M., Reiche, K., Blume, A. & Garidel, P. Viscosity measurements of antibody solutions by photon correlation spectroscopy: An indirect approach—limitations and applicability for high-concentration liquid protein solutions. *Pharmaceutical development and technology* **18**, doi:<https://doi.org/10.3109/10837450.2011.649851> (2012).

Ref 48: Pikaev, A. & Kartasheva, L. Radiolysis of aqueous solutions of ethylene glycol. *International Journal for Radiation Physics and Chemistry* **7**, 395-415, doi:[https://doi.org/10.1016/0020-7055\(75\)90079-0](https://doi.org/10.1016/0020-7055(75)90079-0) (1975).

4. Figure 1 a and b present G-values that change with absorbed dose. Normally, G-values would describe the yields at very low conversions but here I assume that the G-value describes the total change at a given dose. Please discuss why the presented G-values are changing with dose from a mechanistic point of view.

We have clarified G-values for “specified dose” in figure 1 caption.

Regarding why G-values change with increased absorbed doses, it depends on analyte and radiation type in question. For γ -ray irradiated acetol production, the decrease in trend is due to a conflicting reaction from solvated electrons upon acetol. A new paragraph on page 11 explains this decrease in trend:

“Lastly, the solvated electron (e_{solv}^-) being fairly unreactive towards alcohols is however fairly active towards carbonyls groups²⁷ such as in acetol. Therefore, the decrease of γ -ray acetol G-values with increased dose as indicated in Fig. 1a is explained due to the conflicting reaction in Fig. S2f.”

For mixed-field neutron + γ -ray acetol production, the increasing trend seen in Fig.1 can be explained by through the examination of the diffusion-based acetol mechanism. This diffusion depends on two factors of temperature and viscosity (page XX):

“The additional heating effect caused by increased absorbed doses would raise the temperature of the solution, lower viscosities, increase diffusion constants and increase the reaction rates for diffusion-controlled reactions, such as those for acetol and solketal. For water radiolysis, there is an estimated 24°C temperature increase for 100 kGy absorbed dose²⁷, compared with a 4.8°C rise at 20 kGy. Thermal energy would be able to dissipate for longer γ -only irradiations, whereas for higher-dose rate mixed-field irradiations, the thermal energy would accumulate. These two sample factors explain; i) the increase in G-values seen from neat to diluted glycerol samples, and ii) increasing acetol G-values with absorbed dose of mixed-field neutron+ γ -ray irradiations (Fig. 1a).

For the increase in G-value from both γ -ray and mixed-field for solketal production, that is now covered in page XX. This also overlaps with the response to comment #5 (Underlined is the relevant text):

“A similar dependence on the short-lived acidic species is hypothesised here for ketalization reaction to form solketal, provided sufficient availability of acetone in the starting mixture, as indicated in Fig. 4g. The reaction proceeds chemically, with the necessary radical combinations being highly improbable. Acetone as a limiting reagent, explains the lower yields (a factor of 10 lower) of solketal compared with acetol in the neat samples. The increase in the G-value trend observed in Fig.1b with absorbed dose is also explained by the accumulation of acetone for the solketal reaction for higher doses.”

5. The G-values for solketal are one order of magnitude lower than the G-values for acetol under the same conditions (Figure 1). Please provide an explanation to this (which I assume can be found in a more detailed mechanism).

An interesting point is raised here. We have included a sentence in the altered discussion to explain these G-value differences between acetol and solketal as shown in Fig. 1, specifically involving the limitation of the acetone reagent for solketal. Following is an additional paragraph on page 10 (Underlined is the relevant text):

“A similar dependence on the short-lived acidic species is hypothesised here for ketalization reaction to form solketal, provided sufficient availability of acetone in the starting mixture, as indicated in Fig. 4g. The reaction proceeds chemically, with the necessary radical combinations being highly improbable. Acetone as a limiting reagent, explains the lower yields (a factor of 10 lower) of solketal compared with acetol in the neat samples. The increase in G-value trend observed in Fig.1b with absorbed dose is also explained by the accumulation of acetone for the solketal reaction for higher doses”.

6. Why do the G-values depend on dose rate?

The reviewer raises an important query that we have carefully considered and expanded upon with regards to the known spur diffusion theory. We refer to Fig. 3, Fig. 4 and Fig. S2 and the following modified section on page 10, (the underlined text is new):

“The second radiation factor relates to dose rate which is linked⁴³ to the yields of the reactive species such as the stabilised glycerol cation or H_3O^+ that participate in diffusion-controlled mechanisms. The higher dose rate observed for mixed fields compared to γ radiations would be similar to increasing LET⁴⁵, due to the increased probability of spurs overlapping and higher recombination rates of the reactive species. Diffusion-controlled acetol and solketal mechanisms requiring such species would be limited which is supported by the data in Fig. 1c,d and Fig. 2. This is supported by the dose-rate dependence for solketal data in Fig 1b.”

We have introduced two references that describe spur diffusion theory:

Ref 36: Mozumder, A. & Magee, J. L. A Simplified Approach to Diffusion-Controlled Radical Reactions in the Tracks of Ionizing Radiations. *Radiation Research* **28**, 215-231, doi:<https://doi.org/10.2307/3572191> (1966)

Ref 37: Ganguly, A. & Magee, J. Theory of radiation chemistry. III. Radical reaction mechanism in the tracks of ionizing radiations. *The Journal of Chemical Physics* **25**, 129-134, doi:<https://doi.org/10.1063/1.1742803> (1956).

7. Figure 2 shows some very interesting results for mixtures of glycerol and water and for mixtures of glycerol, water and acetone. It is clear that for the glycerol/water mixture (gamma irradiated), the yield of acetol at a given dose decreases with decreasing glycerol concentration. In the glycerol/water/acetone mixture, this is not the case. In this case, the yield of solketal is also very high (and independent of glycerol concentration). These observations should be more deeply discussed as they most certainly can prove mechanistic insights. Why is the yield of acetol lower in the ternary mixtures than in the binary mixtures? Competition kinetics? Is acetol converted into solketal?

We agree with the reviewer in that competition kinetics between acetol and solketal mechanisms are occurring within the ternary mixtures – specifically for a catalytic acidic species for their respective mechanisms. We expand upon this in the following section of the discussion (Page 11):

“For the ternary mixtures in Fig. 2, with acetone no longer a limiting reagent for solketal production, there would competition kinetics for the short-lived acidic species between acetol and solketal processes shown in Fig. 4 reducing acetol *G*-values compared to binary mixtures.”

8. When reading the manuscript, it is easy to get the impression that the formation of acetol and solketal can be attributed to acidification induced by radiation. If this would be the case, a reference experiment in buffered solution should be conducted in order to demonstrate this. Considering the very high concentrations of glycerol, I am surprised to see that direct radiolysis of glycerol is not discussed. Considering the high reactivity of both hydroxyl radicals and hydrogen atoms towards glycerol and acetone, I am also very surprised not to see any discussion on these possible reactions. These points must be addressed.

We agree with the reviewer’s comments, direct action upon glycerol has been expanded upon as referred to in the previous comment 3. As indicated by new Fig. 3 and 4, hydroxyl radicals and hydrogen radicals do initiate a homolytic hydrogen-abstraction process from glycerol for acetol production but play no role in solketal production. With regards to carbonyl groups such as present in acetol or acetone, the solvated electron (e_{solv}^-) triggers a conflicting reaction, as now indicated in Fig S2f and described in the added text (Page 11):

“Lastly, the solvated electron (e_{solv}^-) being fairly unreactive towards alcohols is however fairly active towards carbonyls groups²⁷ such as in acetol. Therefore, the decrease of γ -ray acetol *G*-values with increased dose as indicated in Fig. 1a is explained due to the conflicting reaction in Fig. S2f. For the ternary mixtures in Fig. 2, with acetone no longer a limiting reagent for solketal production, there would competition kinetics for the short-lived acidic species between acetol and solketal processes shown in Fig. 4. reducing acetol *G*-values compared to binary mixtures.”

9. As mentioned above, the safety aspects of the proposed nuclear-driven production of renewable fuel additives are not mentioned. In general, introducing even the slightest modification in a nuclear facility requires thorough safety analysis. The proposed facility making use of the nuclear reactor itself or a storage pool for spent nuclear fuel would involve a major change in the nuclear facility with risks that are quite obvious in terms of risk for corrosion, leakage, contamination, fire, etc. In addition, it is not clear how the integration of the fuel additive production facility and the nuclear power plant will affect the performance of the nuclear power plant. Since safety is such a crucial part of nuclear technology, I would strongly recommend the authors to include a paragraph addressing the safety issues.

The authors recognise the concerns about safety about facility modifications. We have conceded that further studies need to be done regarding process modelling, corrosion and safety. We have discussed the co-generation integration suitability further in the new discussion section of the paper, titled, “Immediate suitability of the scheme for integration with nuclear facilities “on (Page 14 and 15):

Due to the benefit of the γ -only scheme, we foresee no future direct integration with an operating reactor and consequently no related reactor performance issues. Instead, we propose the utilization of spent fuel storage facilities the consideration of nuclear co-generation. Additionally, given the existing regulatory compliances of other operating processes at commercial scale, i.e. spent fuel reprocessing, the hazards involved in the inclusion of a physically separated co-generation system would be markedly reduced comparatively.

Reviewer #2 (Remarks to the Author):

The manuscript devoted to the combination of technologies using ionizing radiation for the deposition of valuable raw materials from spent glycerin will certainly be of interest to specialists in these fields of science.

Reducing the amount of greenhouse emissions using the energy of the “peaceful atom” is a very promising direction. The use of a scientific reactor showed the consistency of the considered process.

My recommendation is to determine the composition of unknown 46.5% of the components obtained in the process of radiolysis to assess their safety at various dose rates, since a wide variety of products can be formed, including those that pose a danger to personnel.

We agree that this is a relevant issue that is also raised in comment 9 of Reviewer#1. The detected products via liquid sampling GC-MS have been listed in Table S1 of the supplementary material, including formaldehyde, acetaldehyde, methanol, acetone, etc. The newly revised mechanism explains the formation of these compounds (new Figure 3 a and b) by direct ionization and subsequent ion fragmentations of glycerol. The following sentence was added within the production mechanisms section (Page 7):

“Direct ionization and subsequent ion fragmentations on glycerol are shown in Fig. 3a and b, producing excited radical cations and non-ionizing electrons ($C_3H_8O_3^{*\cdot+}$, e^-). The electrons solvate after $\sim 10^{-10}$ s³⁵, the excited radical cations can react several ways via fragmentations and will contribute to the majority of C-O and C-C cleavages which neutralize to result in the synthesis of the observed smaller compounds such as formaldehyde, acetaldehyde, methanol and acetone.”

Furthermore, we anticipate the production of H₂ gas as the main minor gaseous product as it is the case with the radiolysis of most hydrocarbons and alcohol reported in the literature. This is presented in new Figure 4b and c.

We agree that some of these products can be hazardous and therefore the following sentence was added in the new discussion section on suitability:

“For future nuclear chemical co-generation processes to successfully enacted, research into process modelling, corrosion, and safety aspects will need to be completed.”

In subsequent studies, I would like to see the thermodynamic characteristics of the process under conditions of exposure to ionizing radiation, as well as the analysis of gaseous products of radiolysis, which may call into question the use of this method for processing glycerin in the conditions of power reactors of nuclear power plants. Such initiatives, without proof to the contrary, can lead to emergency modes in the glycerin processing loop, which may affect the stability of the NPP operation. The world is aware of enough sad incidents at the enterprises for the reprocessing of spent nuclear fuel. Given the updated data, the design of the reactor may need to be modernized. Have you performed dosimetry of the irradiated samples?

Many thanks for suggesting to perform further studies on thermodynamic characteristics of the process under conditions of radiation exposure. It will certainly be a part of our future plans. We agree that dosimetry is an important aspect of radiation chemistry. Given the extreme environment of the research reactor (high dose rates and neutron fluences) and the applicability of chemical dosimeters, mixed-field n+ γ dosimetry relied on a validated, accurate MCNP model of the TRIGA reactor as described in the methods. This validated model and research reactor has been regularly utilized as a reference facility (for CEA and CERN). Delayed γ -only irradiations relied on regular reactor power calibrations with known, calibrated ionization chambers as in the method.

The irradiations section of the methodology has been slightly expanded and clarified upon (Page 16-17 within the experimental section). The new text is underlined as follows:

“Using a calibrated ionization chamber, a factor of 14250 Gy s⁻¹ W⁻¹ for triangular irradiation channel was determined with the accuracy of 10%⁵⁸ and was utilized for γ -dose determination. Due to the logarithmic decay of the reactor power (<1.5 kW) and consequently changing dose rate, the average power was taken for short time intervals and the absorbed dose totalled over time until the required dose was achieved.”

Furthermore, the following sentence was modified (underlined) to describe the calculations:

“For mixed-field irradiations during reactor operation, absorbed dose determinations were supported by calculations using the calibrated, validated JSI TRIGA reactor MCNP model 28.59.60 in addition to determining dose uncertainties.”

In terms of safety, we recognise the high import for full safety analysis, examining corrosion and radiation-chemical co-generation designs, as mentioned in the previous answer.

Surprisingly, the presentation of the material is not standard, there are no conclusions. There is no clear Introduction section (“must begin with the heading “Introduction””), no clear Results/Conclusions section (“must begin with the heading “Results””). Also, there is only one Table in the manuscript text, but it is named “Table 2” (should be Table 1 / Table). As a result, the article can be accepted for publication with minor revision.

As suggested, the format of the article was changed in such a way to include an introduction, results, discussion and conclusion sections, all aligned with the Communications Chemistry formatting. “Table 2” has been changed to “Table 1”.

Reviewer #3 (Remarks to the Author):

The manuscript develops the argument that use of radiation from nuclear reactors to convert mixtures of glycerol and acetone to higher value products including acetol and solketal might improve the economics of nuclear power.

The paper presents radiochemical data with G values from irradiations of glycerol +/- water/acetone which appear to be carefully obtained with mechanistically sensible explanations.

Although addressing the cost of nuclear power and improving the economic value proposition is extremely important, and the use of otherwise un-utilized radiation from nuclear reactions might help in improving the economics, how to make use of radiation remains an open question. The nuclear radiation must replace something expensive in an existing chemical process in order to offer some cost advantage. Unfortunately, the authors have selected chemical processes that can already be performed under relatively mild conditions and the radiation is used to replace the low cost acidic catalysts already used, which offers little savings.

It will almost never make sense to make chemical products that are done in low capital low energy existing processes, it would be highly recommended that reactions with large energetic barriers or where expensive non-selective pathways are presently used be examined to provide clear incentives for use of nuclear technology.

Issues requiring attention from the authors:

1) The authors indicate glycerol as having zero cost, in fact, although early in the history of biodiesel the market was saturated and the price very low, global demand increased with new uses for glycerol and demand has outpaced supply. Prices now are approximately \$200/ton for crude glycerol and higher when refined. The authors must address the impact of feed cost on their conclusions. If radiochemistry is to be used it will be important to know the impact of impurities typical from the unrefined crude feedstock.

Thank you for the suggestion. We have performed additional research on the prices of crude and refined glycerol from more current literature. The following section has been included at the end of page 2 and the beginning of page 3 (new text underlined):

“Coupling nuclear and biorefinery processes. Glycerol is produced as a by-product from biodiesel production but also has potential as a low-value source of valuable, renewable chemicals. Since the saturation of the glycerol market in 2006, due to the increase in biodiesel production^{12,13}, the price of glycerol whilst remaining relatively low has been rising steadily. Historically, glycerol has been unusable in high-value applications^{14,15} with thousands of tonnes of crude glycerol being disposed of at negative prices in 2014¹⁶. As of 2017 in the EU, prices of crude and refined glycerol are at 200-300 € per tonne¹⁷ and 500-700 € per tonne (pre-pandemic), respectively. With glycerol production expected to triple by 2030¹⁸ and oversupply expected to continue, deriving useful feedstocks from waste glycerol is important if biodiesel production is to be sustainable.”

The following references have been added which refer to the glycerol market:

Ref 13: Quispe, C. A. G., Coronado, C. J. R. & Carvalho Jr, J. A. Glycerol: Production, consumption, prices, characterization and new trends in combustion. *Renewable and Sustainable Energy Reviews* **27**, 475-493, doi:<https://doi.org/10.1016/j.rser.2013.06.017> (2013).

Ref 16: Pagliaro, M. in *Glycerol: The Renewable Platform Chemical* (ed Mario Pagliaro) 1-21 (Elsevier, 2017).

Ref 17: ICIS. *Spot Europe glycerine prices surge on limited supply, healthy demand*, <<https://www.icis.com/explore/resources/news/2017/03/23/10090697/spot-europe-glycerine-prices-surge-on-limited-supply-healthy-demand/%3e> (2017).

2) Reference should be made to the many papers with thermochemical pathways and simple acid catalysts to produce acetol from glycerol. Also, the techno-economic analysis by Al-Saadi and colleagues for acid catalysed production of Solketal from acetone + glycerol, Front Chem. 2019; 7: 882.

The value of solketal ~ \$3/kg should be critically questioned.

We thank the reviewer for suggesting an additional resource associated with techno-economic analysis. We have added the following paragraph (new text underlined) to support the statements about the solketal prices, citing Al-Saadi et al. 2019 (Ref. 51) towards the end of the discussion section (Page 14):

“This would alleviate forecast demand for biomass-derived, sustainable fuel additives which are expensive. Notwithstanding the scarcely available data⁵¹ for solketal prices (≈3000 \$ per tonne), it could be produced via irradiations at lower costs than suggested chemoselective methods (≈2,088 \$ per tonne). This would be due to negligible radiation processing costs²⁷ from spent fuel sources if process yields and selectivity can be improved to match chemoselective methods. However, solketal production costs would depend on feedstock prices, a nuclear-biorefinery plant concept would avoid feedstock purchasing market issues and costs since both biodiesel and solketal could be produced on the same site.”

Also, we have included additional references about thermochemical pathways to produce acetol from glycerol (Ref. 24, 25 and 26):

Ref 19: El Roz, A., Fongarland, P. & Capron, M. Glycerol to Glyceraldehyde Oxidation Reaction Over Pt-Based Catalysts Under Base-Free Conditions. *Frontiers in Chemistry* **7**, 156, doi:<https://doi.org/10.3389/fchem.2019.00156> (2019).

Ref 24: Montes, V. *et al.* Synthesis of different ZnO-supported metal systems through microemulsion technique and application to catalytic transformation of glycerol to acetol and 1,2-propanediol. *Catalysis Today* **223**, 129-137, doi:<https://doi.org/10.1016/j.cattod.2013.09.021> (2014).

Ref 25: Kinage, A. K., Upare, P. P., Kasinathan, P., Hwang, Y. K. & Chang, J.-S. Selective conversion of glycerol to acetol over sodium-doped metal oxide catalysts. *Catalysis Communications* **11**, 620-623, doi:<https://doi.org/10.1016/j.catcom.2010.01.008> (2010).

Ref 26: Corrêa, I., Faria, R. P. V. & Rodrigues, A. E. Continuous Valorization of Glycerol into Solketal: Recent Advances on Catalysts, Processes, and Industrial Perspectives. *Sustainable Chemistry* **2**, 286-324, doi:<https://doi.org/10.3390/suschem2020017> (2021).

The following text citing these references is added at the end of the introduction section to address thermochemical pathways (Page 3):

“While chemoselective advances for acetol^{24,25} and solketal²⁶ have been reported, radiation-initiated processing has not been explored for glycerol and could offer several advantages²⁷: i) catalytic deactivation or poisoning is not a concern; ii) reactions can proceed at ambient temperatures and pressures; and iii) the availability of irradiating large reaction volumes due to the penetrating power of ionizing radiation; and iv) the utilization of waste ionizing sources (spent fuel rods) would also result in negligible radiation-related processing costs.”

3) The data presented indicates that gamma radiation is the most effective for the product selectivity and G's. This is also true because neutron irradiation of feeds with lots of impurities might activate impurities rendering the products radioactive and valueless. This point should be made very clear in the paper.

We agree with the reviewer's comment, especially for crude glycerol, the following sentence has been included in the discussion section which highlights the benefit of γ -only irradiations (Page 13):

“An additional advantage, γ -only irradiations of a spent fuel pool remove the potential of radioactive activation of metal impurities, especially when high-sodium, crude glycerol could be considered as a feedstock.”

4) Although, it is always nice to see Monte Carlo radiation transport models, the value for this paper is unclear. Moving most all of that material to the supplementary information would make the major messages more clear.

As suggested by the Reviewer, two MCNP images have been moved from Fig. 3c and d to Fig. S3a and b in the supplementary material.

5) Throughout the paper editing is needed – I refer that chore to the editors.

As identified, the updated manuscript has been edited by the authors towards the formatting requirements of the Communications Chemistry journal. We have carefully reviewed the editing. As a result, we have included a couple of editing changes (highlighted in green). Hopefully to minimize future editorial changes.

REVIEWERS' COMMENTS:

Reviewer #1 (Remarks to the Author):

The authors have acted upon my previous comments satisfactorily and I recommend acceptance for publication.

Reviewer #2 (Remarks to the Author):

The authors have significantly revised the material presented in the article. The mechanism of decomposition of pure glycerin, as well as its various solutions, has been revised. The radiation-chemical yields of the main products have been specified. A section, «Immediate suitability of the scheme for integration with nuclear facilities», has been added, which addresses the topic of safe reprocessing of spent nuclear fuel. The results presented in the article are undoubtedly interesting from a scientific point of view. However, they are far from practical application. Nevertheless, the article can be accepted for publication.

Reviewer #3 (Remarks to the Author):

The authors have satisfactorily addressed the comments from the reviews and the revised manuscript is appropriate for publication.

Response to Reviewers' Comments (09/08/2021)

Communications Chemistry – Nature Paper ID: COMMSCHEM-21-0127B

Text in red – Reviewers' comments

Text in black – Authors' response

Reviewer #1 (Remarks to the Authors):

The authors have acted upon my previous comments satisfactorily and I recommend acceptance for publication.

Reviewer #2 (Remarks to the Author):

The authors have significantly revised the material presented in the article. The mechanism of decomposition of pure glycerin, as well as its various solutions, has been revised. The radiation-chemical yields of the main products have been specified. A section, «Immediate suitability of the scheme for integration with nuclear facilities», has been added, which addresses the topic of safe reprocessing of spent nuclear fuel. The results presented in the article are undoubtedly interesting from a scientific point of view. However, they are far from practical application.

Nevertheless, the article can be accepted for publication.

Reviewer #3 (Remarks to the Author):

The authors have satisfactorily addressed the comments from the reviews and the revised manuscript is appropriate for publication.

The authors would like to thank the reviewers for their time and valuable feedback, their input has been important to improve this research and article as a whole.